# Assessing emergence risk of double-resistant and triple-resistant genotypes of *Plasmodium falciparum*

Eric Zhewen Li [1], Tran Dang Nguyen [1], Thu Nguyen-Anh Tran [1], Robert J. Zupko [1] & Maciej F. Boni [1,2] ✉

Delaying and slowing antimalarial drug resistance evolution is a priority for malaria-endemic countries. Until novel therapies become available, the mainstay of antimalarial treatment will continue to be artemisinin-based combination therapy (ACT). Deployment of different ACTs can be optimized to minimize evolutionary pressure for drug resistance by deploying them as a set of co-equal multiple first-line therapies (MFT) rather than rotating therapies in and out of use. Here, we consider one potential detriment of MFT policies, namely, that the simultaneous deployment of multiple ACTs could drive the evolution of different resistance alleles concurrently and that these resistance alleles could then be brought together by recombination into double-resistant or triple-resistant parasites. Using an individual-based model, we compare MFT and cycling policies in malaria transmission settings ranging from 0.1% to 50% prevalence. We define a total risk measure for multi-drug resistance (MDR) by summing the area under the genotype-frequency curves (AUC) of double- and triple-resistant genotypes. When prevalence ≥ 1%, total MDR risk ranges from statistically similar to 80% lower under MFT policies than under cycling policies, irrespective of whether resistance is imported or emerges de novo. At 0.1% prevalence, there is little statistical difference in MDR risk between MFT and cycling.

Malaria eradication efforts have accelerated over the past twenty years with increased global funding, lower malaria transmission in many regions, and 23 previously endemic countries achieving zero indigenous malaria cases for three consecutive years[1]. One major reason for these prevalence reductions and accelerated paths to elimination is the widespread adoption over the past two decades of artemisinin combination therapies (ACTs) which have maintained ~95% treatment efficacy in uncomplicated *Plasmodium falciparum* infections in most countries for much of this time period[2,3]. However, as artemisinin-resistant genotypes have now emerged at least six times on three continents[4–12], a number of countries—Cambodia, Myanmar, Vietnam, Laos, Papua-New Guinea, Rwanda, Uganda—face the prospect of lower

ACT cure rates for the foreseeable future and continually increasing drug-resistance trends. Historically, the spread of drug-resistance is not a self-limiting process[13,14]. Artemisinin-resistant genotypes are likely to spread geographically and undermine malaria elimination efforts worldwide.

One straightforward approach to slowing down drug-resistance evolution is to force malaria parasites into a constantly changing treatment environment with multiple simultaneous or near-simultaneous therapeutic challenges[15]. Combination therapy achieves this goal, and triple therapies are currently being trialed[16,17]. In the absence of triple or quadruple therapies, simultaneous deployment of multiple first-line therapies (MFT) is the next best option as this forces

[1]Center for Infectious Disease Dynamics, Department of Biology, Pennsylvania State University, University Park, PA, USA. [2]Centre for Tropical Medicine and Global Health, Nuffield Department of Medicine, University of Oxford, Oxford, UK. ✉e-mail: mboni@temple.edu

the parasite population to encounter a different ACT every several weeks[18]. A third option is drug cycling or rotation policies, but MFT tends to outperform cycling policies[19,20] as it creates higher environmental variability (more drugs seen per unit time) thus slowing the pace of parasite adaptation. One potential concern with MFT strategies is the potential risk of generating multi-drug resistant (MDR) genotypes earlier than expected, as the concurrent deployment of multiple therapies puts selection pressure on different alleles at the same time[21]. These different resistance alleles could be brought together into MDR genotypes via recombination or successive de novo mutation. Population-genetic theory predicts that increased rates of recombination may speed up the time of appearance of these MDR genotypes[22,23], but neither the current risk nor the relative risks under different drug deployment strategies are known.

In this study, we quantify the risk of double-resistant and triple-resistant genotypes emerging under an MFT deployment, and compare this to the risk under two different cycling policies—a five-year cycling policy and the status quo policy of rotating out a therapy after 10% treatment failure is observed. We assume that the three most commonly used ACTs—artemether-lumefantrine (AL), artesunate-amodiaquine (ASAQ), and dihydroartemisinin-piperaquine (DHA-PPQ)—are available for use. And, we define MDR risk as the area under the genotype-frequency curve for a particular MDR genotype or across all MDR genotypes. Evaluations are performed with a previously calibrated individual-based simulation of *Plasmodium falciparum* transmission and drug-resistance evolution[20,24]. Our starting hypothesis is that because MFT selects for multiple resistant genotypes simultaneously, we will see earlier emergence of double- and triple-resistant genotypes under MFT due to the action of recombination. The major scientific advance in the following analysis is the characterization of rare evolutionary events through a large-scale simulation approach.

## Results

National malaria control programs (NMCP) have several choices for the deployment of different artemisinin combination therapies. The adopted status quo management of the last two decades of ACT use has been an 'adaptive cycling' policy where therapies are rotated out when a 10% treatment failure threshold is reached per WHO recommendation;[25] this drug switch typically takes one to three years to implement, and we model this as a one-year delay in the present analysis. A second approach, adopted officially by thirteen countries, is an MFT deployment where multiple ACTs are recommended as co-equal therapies[1]. A third commonly discussed approach is ACT cycling on a pre-set schedule of five years (or similar) something that has not yet been adopted by any NMCP. We model these three drug-deployment approaches on a 20-year time scale in five different prevalence scenarios (PfPR$_{2-10}$ = 0.1%, 1%, 5%, 25%, 50%) and three drug coverage settings (20%, 40%, 60%) in order to determine which approaches are associated with the lowest or highest risk of emergence of multi-drug resistant genotypes.

### Emergence of double- and triple-resistance in specific scenarios
As expected from standard evolutionary theory, when multiple types of antimalarial therapies are deployed, multi-drug resistance does emerge and its onset comes at a delay from the arrival of single-resistant genotypes. In the present analysis, double-resistants refer to genotypes that are resistant to one artemisinin derivative and one partner drug while triple-resistants refer to genotypes that carry resistance to artemisinin and two partner drugs. Table 1 shows the five 'maximally resistant' double-resistants and triple-resistants that we track in this analysis. These genotypes are maximally resistant only in the framework of the four loci and two copy-number variants that we consider here. The modeled trajectories of drug resistance in Fig. 1A, B show that the time to 0.01 genotype frequency of an artemisinin-piperaquine-resistant double-mutant is 7.1 years (95% range:

5.3y–13.3 y) when DHA-PPQ is deployed as first-line therapy in a status-quo adaptive cycling framework (and under a particular parameterization of the mutation rate[26]); treatment coverage is 40% in this scenario and PfPR$_{2-10}$ = 5%. The 0.01 frequency threshold was chosen as this is a high enough level that drug-resistance is not susceptible to random extinction, and low enough that the major contributory drivers in getting to this point are the genetic processes (mutation, recombination) responsible for the appearance of this mutant. When ASAQ is used as first-line therapy, the median time to 0.01 frequency of the maximally-resistant ASAQ double-resistant (Table 1) is 8.2 years (95% range: 5.7y–11.1 y). While the double-resistants emerge on a time scale of 5 to 13 years, single-resistant genotypes (blue lines, Fig. 1A, C) rise to high frequencies in several years or may already be present in settings with high levels of pre-existing partner-drug resistance. These two scenarios show typical patterns of MDR evolution and provide useful outcome measures to track when attempting to optimize a policy to delay or minimize the negative clinical and public health effects of MDR malaria. One measure we use here is the time until 0.01 MDR genotype frequency ($T_{.01}$, black dots and triangles, Fig. 1) and the second measure is the total area under the MDR frequency curve (AUC) which corresponds to the total number of risk-days of MDR circulation.

When artemether-lumefantrine (AL) is deployed, maximum double resistance (Table 1, row 5) appears much more slowly than under ASAQ or DHA-PPQ for three reasons: (1) AL resistance involves more mutational steps, with some pathways disallowed since a genotype with two copies of the *pfmdr1* gene is not allowed to acquire two independent mutations to change both copies simultaneously, (2) DHA-PPQ efficacy drops to lower levels than the lowest efficacies for AL and ASAQ, making DHA-PPQ resistance evolution faster, and (3) the simulations are started with some AQ resistance (for historical accuracy). A previous analysis showed that slower lumefantrine resistance evolution is likely a general property of falciparum resistance evolution and that AL is the most likely among the ACTs to select for artemisinin resistance first and partner-drug resistance second (Supplementary Fig. 19 in Watson et al.[26]). Figures 1F and 1H show no trajectories (out of 100 runs) of the AL maximal double-resistant reaching 0.01 genotype frequency after 20 years. Nevertheless, weaker double resistance to AL does emerge in 7.6 years (95% range: 5.5y–16.7 y) (Fig. 1E) and 8.2 years (IQR: 5.5y–16.4 y) (Fig. 1G).

A comparison at 5% PfPR$_{2-10}$ and 40% drug coverage shows that emergence of double- and triple-resistance tends to occur later under MFT than under cycling policies, despite the possibility of recombination (hypothesized to be higher under MFT) bringing together two or three different resistance alleles into a single MDR genotype (Fig. 2). Specifically, the triple-resistant to DHA-PPQ and amodiaquine emerges later under MFT (median time 17.1 years) than under either cycling policy (median times 16.5 years and 8.4 years; both Mann-Whitney $p < 0.01$). Likewise, the double-resistant to ASAQ emerges later under MFT (median time 12.6 years, versus median times of 9.4 and 8.3 years for cycling policies; both $p < 10^{-9}$). The double-resistant to DHA-PPQ emerges slightly earlier under MFT (14.9 years) than under 5-year cycling (15.5 years; $p = 0.152$) but much later than under adaptive cycling (7.2 years; $p < 10^{-33}$). The full 20-year outlook is still worse for 5-year cycling than MFT because the slightly delayed emergence of double resistance is followed by rapid exponential growth of this genotype. The slower spread of drug resistance is characteristic of MFT policies[19,20,27–29], and this is the reason that the earlier emergence of double resistance under MFT in this scenario poses a lower comparative threat, because the fixation of the double-resistant genotype is slower post-emergence. Full double-resistance to AL and triple-resistance to AL and PPQ do not emerge in this epidemiological scenario, but the second and fifth rows of Fig. 2 show graphically that MFT is associated with equal or lower MDR risk for these two genotypes when compared to cycling policies.

**Table 1 | List of five maximally-resistant genotypes tracked in this analysis**

| Name | Genotype | Treatment Failure |
|------|----------|-------------------|
| DHA-PPQ, AQ triple-resistant | 580Y, 76T, 86Y, Y184, PPQ-resistance present in *pfcrt* | 58.5% for DHA-PPQ 26.5% for ASAQ 9.2% to 20.5% for AL |
| DHA-PPQ, LUM triple-resistant | 580Y, K76, N86, 184F, PPQ-resistance present in *pfcrt* | 58.5% for DHA-PPQ 5.2% for ASAQ 27.7% to 43.0% for AL |
| DHA-PPQ double-resistant | 580Y, PPQ-resistance present in *pfcrt* | 58.5% for DHA-PPQ |
| ASAQ double-resistant | 580Y, 76T, 86Y, Y184 | 26.5% for ASAQ |
| AL double-resistant | 580Y, K76, N86, 184F | 27.7% to 43.0% for AL |

For all five genotypes, *pfmdr1* copy number can be single or multiple.

The overall MDR risk in these 20-year scenarios can be summed up by an AUC measure that counts up the total number MDR risk-days for each maximally-resistant MDR genotype. For multiple first-line therapies, median AUC values are between 22% and 90% lower than AUC values associated with cycling policies (Fig. 2). When looking at the AL double-resistant genotype (Fig. 2, bottom row) and the triple resistant to AL and piperaquine (Fig. 2, second row), these genotypes have median emergence times that are longer than 20 years (with our parameterization of mutation rate) and the median AUC values are zero for all three policies.

In a low transmission scenario with PfPR$_{2-10}$ = 0.1% (Fig. 3) emergence times are long, and risk can be evaluated by determining what percentage of runs carry any MDR risk at all. The maximally-resistant genotypes in this transmission setting do not reach frequencies above 0.01 because the speed of emergence (which is driven mainly by the mutation rate) depends on the absolute number of treated parasite-positive individuals in the population. At low prevalence, there are fewer infections, fewer parasites, and fewer opportunities for mutation. Note that low-transmission regions are generally viewed as exerting stronger selection pressure for drug resistance (due to several different immunity and treatment factors, summarized previously[30–33]), but low-transmission regions also have a low rate of mutant appearance due to low absolute parasite population sizes. Visually, the inter-quartile ranges of all simulation outputs show that the adaptive cycling policy has the highest probability of generating early emergence and non-zero frequencies of multi-drug resistance. Total MDR risk during the 20-year period (measured by AUC) is zero or near-zero for most scenarios, but note that the total risk generated by the ASAQ double-resistant is slightly higher under MFT than under either cycling policy (Fig. 3, fourth row). In this specific case, because (1) the scenarios start with the amodiaquine-resistant *pfcrt* 76T and *pfmdr1* Y184 alleles fixed in the population, and (2) the deployment of MFT is non-adaptive, MFT underperforms the cycling policies at managing emergence risk of this particular genotype. Under MFT, ASAQ is deployed for 33% of treatments despite the presence of high levels of AQ-resistance, but under the 5-year cycling policy ASAQ is used for only 25% of the policy period and the adaptive cycling strategy uses ASAQ for only 10.8% (IQR: 1.62% – 26.6%) of treatments. If we replace the traditional 3-therapy MFT with an MFT deployment of only AL and DHA-PPQ, the total risk generated by the ASAQ double-resistant (median AUC = 0.88, IQR: 0.40–3.28) is lowest under MFT. In general, if partner-drug resistance is pre-existing, a drug deployment policy should account for this and be readily adapted to a version deploying mainly therapies/drugs to which there is currently little resistance (see Supplementary Section 1). An off-the-shelf MFT or cycling policy is not guaranteed to be best in all situations if it is not adapted to current partner-drug resistance conditions.

## Total multi-drug resistance risk across all scenarios

Summing across all five MDR genotypes (Table 1), we can compare total multi-drug resistance risk in different epidemiological scenarios (Fig. 4). Across scenarios with PfPR$_{2-10}$ ≥ 1%, MFT is associated with equal or lower total MDR risk, with MFT's median number of MDR risk-days (AUC) ranging from statistically similar to 79% lower when compared to cycling policies. Under low transmission (PfPR$_{2-10}$ = 0.1%), the total number of malaria cases and the probability of mutation generating a new resistant genotype are both low. Here, MDR risk is generally low for both MFT and cycling policies, with little statistical difference among the approaches. One notable exception occurs at PfPR$_{2-10}$ = 0.1% and 60% coverage where the median AUC values for MFT (0.23), adaptive cycling (0.10; Mann-Whitney $p$ = 0.036), and 5-year cycling (0.13; $p$ = 0.068) show that MFT is associated with the highest MDR risk in this scenario. However, all three of these scenarios show less than one full day of MDR risk over a 20-year period. To remove the effect that transmission setting has on the number of mutations that can be generated, these scenarios were re-evaluated with immigration as the primary source of new mutants, with a Poisson process importing one new mutant per year regardless of transmission setting (Fig. 5). Across all 15 scenarios MFT had equal or lower MDR risk with the number of absolute MDR risk-days under MFT ranging from statistically similar to 80% lower when compared to cycling strategies. Again, at low prevalence (PfPR$_{2-10}$ = 0.1%), there were few statistical differences among the drug deployment strategies.

Note that although MFT minimizes total MDR risk across the three double-resistants and two triple-resistants tracked here, this does not guarantee that MFT minimizes the AUC for each individual genotype. Supplementary Figs. 7 to 34 present the MDR risk profiles for each genotype in each epidemiological scenario, showing that MFT minimizes MDR risk for 86% (but not 100%) of genotype-scenario combinations. Supplementary Table 1 shows a re-evaluation of some of these comparisons if initial MFT deployment is allowed to adapt to current partner-drug conditions.

## Characterization of selection pressure

The major difference between MFT and cycling approaches lies in the types of selection pressures they exert on the diversity of resistant genotypes circulating in the population. Cycling strategies provide a more constant environment with unidirectional selection[19,28] while the MFT environment is more variable and results in more diversifying selection. Figure 6 shows a typical pattern—from the median simulation of a 5-year cycling approach—of unidirectional selection for piperaquine resistance and artemisinin resistance, with each step in the mutational flow clearly showing the acquisition (left to right) of additional mutations conferring resistance to these two drugs. In years 16 to 20, when DHA-PPQ is deployed a second time in a 5-year cycling policy, mutational inflow into the DHA-PPQ-AQ triple-resistant consists of 372 total mutations. Under a median MFT simulation, 318 mutations are observed to this triple-resistant genotype during the same time period. Over the entire 20-year period cycling produces 581 mutations to the triple resistant but MFT produces only 498. This discrepancy in emergence patterns is the likely reason that MDR emerges earlier under cycling policies than under MFT. Supplementary Table 2 shows the number of maximally resistant mutants generated during each five-year time period.

A full characterization of mutation and selection patterns is shown in Fig. 7, over twenty years, for MFT and two variants of 5-year cycling; the order of therapy deployment was reversed in the second cycling

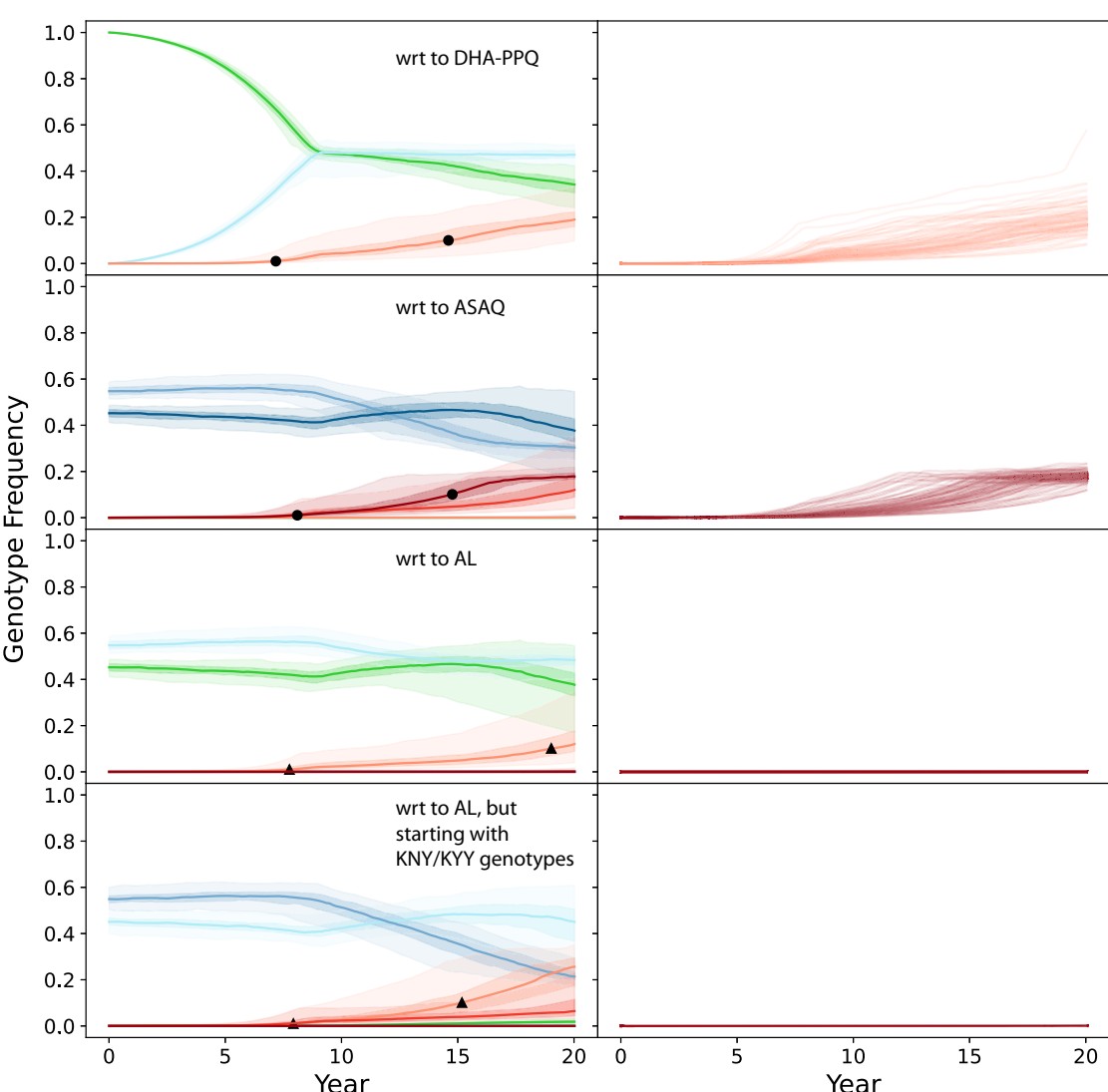

**Median (IQR) genotype frequencies for drug-sensitive (green), single-resistant (blue), and double-resistant (red) genotypes**

**Individual genotypes trajectories of most-resistant double-resistants**

**Fig. 1 | Emergence and evolution of single-resistant and double-resistant P. falciparum genotypes under an adaptive cycling (status quo) drug distribution strategy, where DHA-PPQ is used first, ASAQ is used second, and AL is used last. Epidemiological scenario shown is 5% PfPR$_{2-10}$ and 40% treatment coverage.** Single resistance (blue lines) and double resistance (red lines) are always defined with respect to (**wrt**) a particular artemisinin combination therapy due to the pleiotropic effects of loci in *pfcrt* and *pfmdr1*. Top row shows the evolution of single and double resistance to DHA-PPQ. Second row shows the evolution of single and double resistance to ASAQ. And third and fourth rows show the evolution of single and double resistance to AL. The first three rows show simulations that were started with "TNY" genotypes (76T, N86, Y184) which have some resistance to amodiaquine, while the fourth row shows simulations started with "KNY" genotypes which

show more resistance to lumefantrine. For blue and red lines, the darker the shade the more resistance alleles are present for that resistant genotype. Green lines correspond to genotypes that have no resistance mutations with respect to each ACT. Shaded areas show interquartile ranges from 100 simulations, and light-shaded areas show 90% ranges from 100 simulations. Black circles show the 1% and 10% points for full double-resistants, indicating that full double resistance to DHA-PPQ and ASAQ reaches 0.10 genotype frequency after a median time of approximately 15 years. Black triangles show the 1% and 10% points for any double-mutant double-resistant genotypes to AL, i.e. genotypes with one artemisinin resistance mutation and exactly one lumefantrine resistance mutation. Right-hand panels show the 100 individual trajectories for the full double-resistants, showing that there is substantial variation in the time of emergence for these genotypes.

approach so that therapies were deployed as shortest half-life first and longest half-life last. Visually, the mutation flow diagrams show that MFT, unlike cycling policies, does not produce large 'block flows' of mutation to a single destination genotype. In each 5-year period, the selection regime imposed by MFT produces between 31 and 38 different types of mutants, generating more genotypic diversity than cycling approaches for the majority of each 20-year period. This difference is most pronounced in the first ten years when the diversity of mutants produced by MFT is 1.4 to 3.3 times higher than under either

cycling policy, showing that cycling policies do indeed exert more unidirectional selection than MFT. The absolute numbers of mutations produced by MFT and cycling are approximately equal: 26,930 for MFT, 28,414 for cycling, and 28,616 for reverse cycling.

**Sensitivity analyses**
The greatest uncertainty in projecting the future of multi-drug resistance to artemisinin combination therapies comes from the uncertainty in the fitness changes and epistatic interactions among loci

**Fig. 2 | Evolution of multi-drug resistance under three drug deployment strategies.** Epidemiological scenario shown is 5% PfPR$_{2\text{-}10}$ and 40% treatment coverage. Each row shows the genotype frequency of triple or double resistance to a particular set of antimalarial drugs, with the most resistant genotypes shown in purple (top two rows) or dark red (bottom three rows). In the bottom two rows, medium red corresponds to triple-mutant double-resistance and light red corresponds to double-mutant double-resistance. Median line is shown and interquartile ranges are shaded. No importation is allowed in these figures. Black dots are 0.01 and 0.10 frequency markers for the maximally resistant genotypes. The columns show three different treatment strategies. The outcome measures are the genotype frequency of the maximally-resistant genotype after 20 years ($x_{20}$), the time until the maximally-resistant genotype reaches 0.01 frequency in the population ($T_{.01}$), the total area under the frequency curve of the maximally-resistant genotype (AUC), and the total number of non-discounted treatment failures during the twenty years that a strategy is implemented (NTF). AUC is the most appropriate measure of total MDR risk, and the first, third, and fourth rows show that an MFT strategy generates 27% to 65% less risk than the more optimal cycling strategy. The second and fifth rows show a median value of AUC = 0.0 frequency-days for the maximally-resistant genotypes, for all three drug-distribution strategies.

known to be associated with reduced drug susceptibility. For lumefantrine especially, unambiguous resistant genotypes that do not interact with amodiaquine resistance have yet to be described. A sensitivity analysis on a hypothetical novel lumefantrine resistance locus (Supplementary Figs. 1 and 2) shows that the MDR-risk assessment does not change as MFT is still associated with the lowest risk. Likewise, a sensitivity analysis on the *pfmdr1* Y184F locus and on copy-number variation in *pfmdr1* shows that these two genetic features, with likely weak effects on lumefantrine resistance, do not determine the evolutionary patterns of multi-drug resistance under MFT and cycling strategies (Supplementary Figs. 3 and 4). A general sensitivity analysis shows that treatment coverage has the greatest effect on MDR risk and that longer cycling periods generate more MDR risk even if selection pressure for all resistant phenotypes is not present simultaneously (Supplementary Figs. 5 and 6).

## Discussion

Despite the presence of multi-clonal infections and recombination in certain malaria settings, recombination does not tend to rapidly bring together different types of resistance mutations when multiple first-line antimalarial therapies are deployed in a population. This is important for long-term malaria planning as single-resistant and double-resistant genotypes currently circulate in many regions worldwide[34,35], and two major goals of drug-resistance management this decade will be to ensure that (i) current resistance numbers stay low and (ii) novel resistant types do not emerge. For the near future, all drug-resistance management in malaria will need to be done with the deployment of ACTs (and potentially use of primaquine post-ACT

course[36,37]) as novel therapies will likely not be available until the second half of the decade. This means that management of ACT stocks, continuous molecular surveillance for resistance, and a flexible and adjustable approach to treatment guidelines will all be necessary tools for the WHO and National Malaria Control Programs to successfully minimize drug resistance over the next decade[18]. The challenge with this approach is that ACTs all share an artemisinin component, making successful drug-resistance management difficult—potentially impossible—if artemisinin resistance becomes widespread.

In choosing a specific response plan or drug-resistance management approach at a national scale, our results indicate that NMCPs should begin by considering MFT approaches—and eventually adjustable MFT approaches—as the best option to delay or slow down the emergence and spread of drug-resistant genotypes. An MFT policy creates a more variable environment than cycling approaches, and this helps delay the emergence of resistant genotypes and slows down the spread of these genotypes once they have emerged[19,28]. In addition, as our present analysis shows, despite introducing multiple types of selection pressure which do indeed allow a larger number of resistant genotypes to emerge (Fig. 7), MFT does not cause this parade of genotypes to recombine into novel multi-genic MDR types. This was a major potential concern surrounding MFT policies[15,20,21], but we show here that the total MDR risk under MFT varies from statistically similar to 80% lower than under cycling policies for a range of scenarios with PfPR$_{2\text{-}10} \geq 1\%$.

From these results, the natural conclusion to draw for long-term policy planning in drug-resistance management is that if MFT policies were to be implemented adaptively—in the same way as the status quo

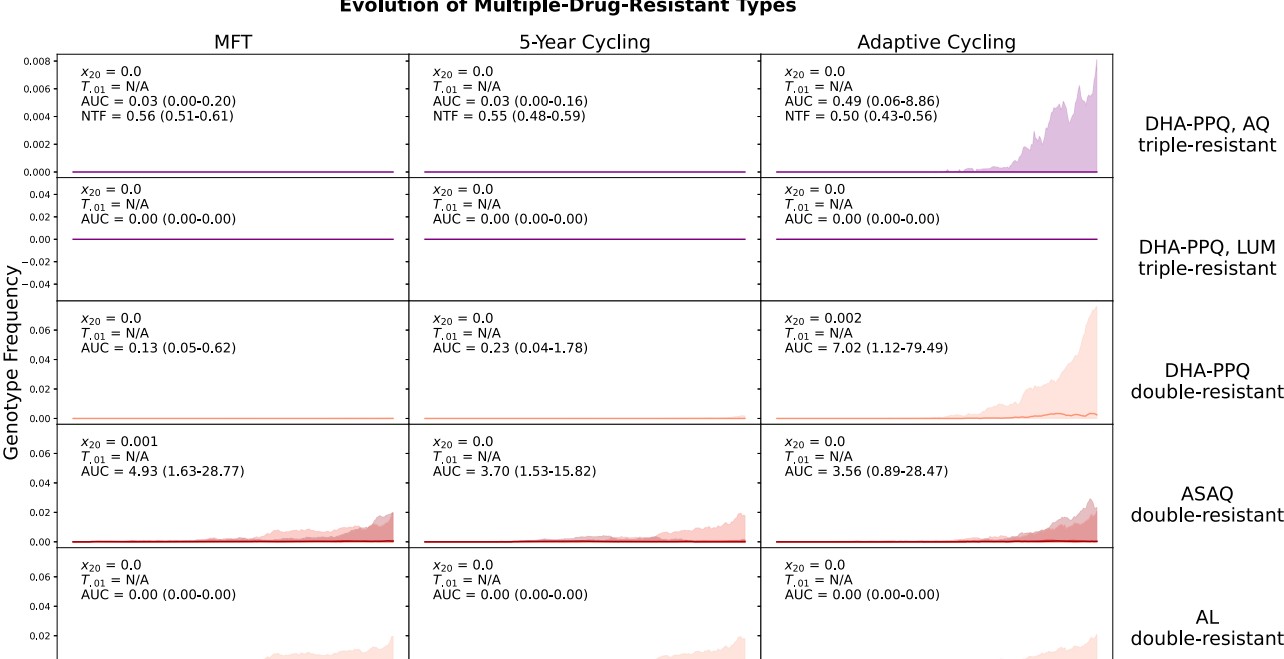

**Fig. 3 | Evolution of multi-drug resistance under three drug deployment strategies.** Epidemiological scenario shown is 0.1% PfPR$_{2-10}$ and 40% treatment coverage. Each row shows the genotype frequency of triple or double resistance to a particular set of antimalarial drugs, with the most resistant genotypes shown in purple (top two rows) or dark red (bottom three rows). In the bottom two rows, medium red corresponds to triple-mutant double-resistance and light red corresponds to double-mutant double-resistance. Median line is shown (nearly always at 0.0) and interquartile ranges are shaded. No importation is allowed in these figures. The columns show three different treatment strategies. The outcome measures are the genotype frequency of the maximally-resistant genotype after 20 years ($x_{20}$), the time until the maximally-resistant genotype reaches 0.01 frequency in the population ($T_{.01}$), the total area under the frequency curve of the maximally-resistant genotype (AUC), and the total number of non-discounted treatment failures during the twenty years that a strategy is implemented (NTF). AUC is the most appropriate measure of total MDR risk, but median AUC = 0.0 for the majority of scenarios in this low-transmission setting. The interquartile ranges in the right-hand column suggest that adaptive cycling has the highest probability of driving maximally-resistant genotypes to high levels. In the first row after 20 years, the triple-resistant reached 0.001 genotype frequency in 32/100 simulations under an adaptive cycling strategy. Under MFT and 5-year cycling, the triple-resistant never rose above 0.001 frequency during the twenty years of the simulation.

is currently adaptive by conducting surveillance and rotating out therapies out at 10% treatment failure—these adaptive MFT approaches would have more favorable profiles than the standard MFT strategies presented here. As an example, there are 11 (non-extinction) comparisons where MFT generates higher risk (when compared to cycling policies) for the ASAQ double-resistant genotype (Supplementary Table 1). The reason is that our simulation scenarios start (as most real-world settings do) with some amodiaquine resistance, and an MFT deployment naively treats one third of patients with ASAQ despite the presence of AQ-resistant genotypes; adaptive cycling policies rotate away from ASAQ when resistance and treatment failure are high but standard MFT does not. Supplementary Table 1 shows that a simple adjustment to a 50/50 MFT deployment of AL and DHA-PPQ resolves this issue and leads to MFT having the lowest AUC for the ASAQ double-resistant, showing that even a minimal effort to adjust an MFT policy to current drug-resistance conditions results in major gains in reducing emergence risk of MDR genotypes.

The simplest adaptation of an MFT policy is to not include, or reduce the frequency of, a particular therapy to which high levels of drug resistance are circulating. In addition to Supplementary Table 1, the analysis in Zupko et al.[38] shows that deployment of AL and DHA-PPQ should be balanced to high levels of AL (for a spatial model parameterized to Burkina Faso's epidemiological parameters) when trying to minimize treatment failure or artemisinin-resistance frequency. In this case, the reason is that PPQ resistance both (i) evolves more rapidly than lumefantrine resistance and (ii) results in higher levels of treatment failure[39]. Essentially, a naïve equal-distribution MFT

approach is not guaranteed to be optimal when one of the therapies has naturally low efficacy or potentially low future efficacy due to high-grade drug resistance.

## Limitations

A simulation approach is necessary for policy evaluations aiming to manage the large-scale epidemiology and evolution of a human pathogen over a years or decades-long time span; a particular strategy's effect on drug-resistance one decade into the future cannot be evaluated with a field trial. Despite their usefulness, simulation approaches come with a number of limitations that need to be made clear to policy makers.

First, the simulations conducted for this analysis (216 epidemiological scenarios with 100 replicates each) are not exhaustive of all possible epidemiological and policy settings that need to be considered; this is impossible for modern simulation approaches where multiple hours or tens of hours are required for each simulation run. Rather, they represent the midpoints and endpoints of ranges of plausible scenarios—e.g. from low transmission to high transmission, or low drug coverage to high drug coverage—but cannot be used to infer how an exact prevalence-coverage setting would fare under a pre-planned but imperfect treatment policy. The purpose of these parameter explorations is to test hypotheses (here, whether MFT generates higher MDR risk) and extract general principles on how and when multi-drug resistance is likely to emerge. Our results show that in this broad range of settings (that includes the majority of malaria-endemic scenarios in Africa) MFT generates less MDR risk than cycling policies.

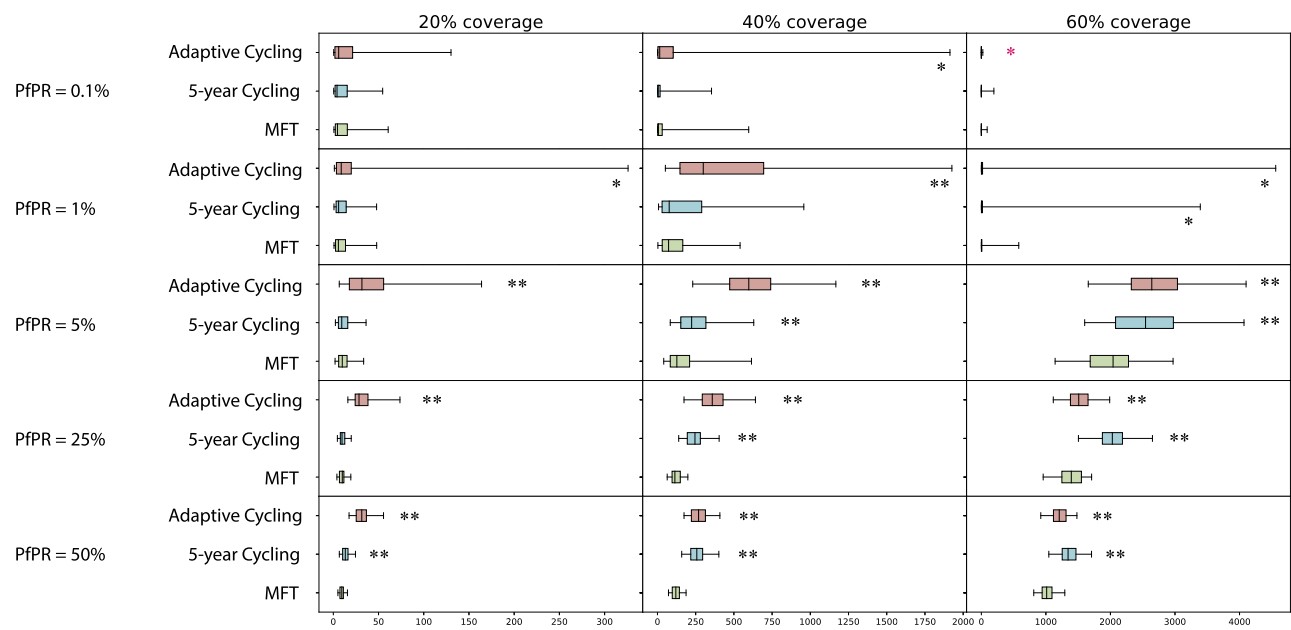

**Fig. 4 | Absolute risk of multi-drug resistance over 20 years.** Each boxplot ($N = 100$ simulations) shows the sum of AUCs across all five maximally-resistant genotypes (Table 1). Boxplot whiskers are 1.5 times the IQR. AUC comparisons between MFT and 5-year cycling and AUC comparisons between MFT and adaptive cycling are assessed with a Mann-Whitney test, and $p$-value markers (testing whether MFT has lower AUC) are placed next to each boxplot with $p < 0.05$ (*) or $p < 10^{-4}$ (**). In the upper-right panel, the red $p$-value marker indicates that MFT (median AUC = 0.23 risk days) has higher AUC than the adaptive cycling strategy (median AUC = 0.10 risk days) with $p = 0.037$.

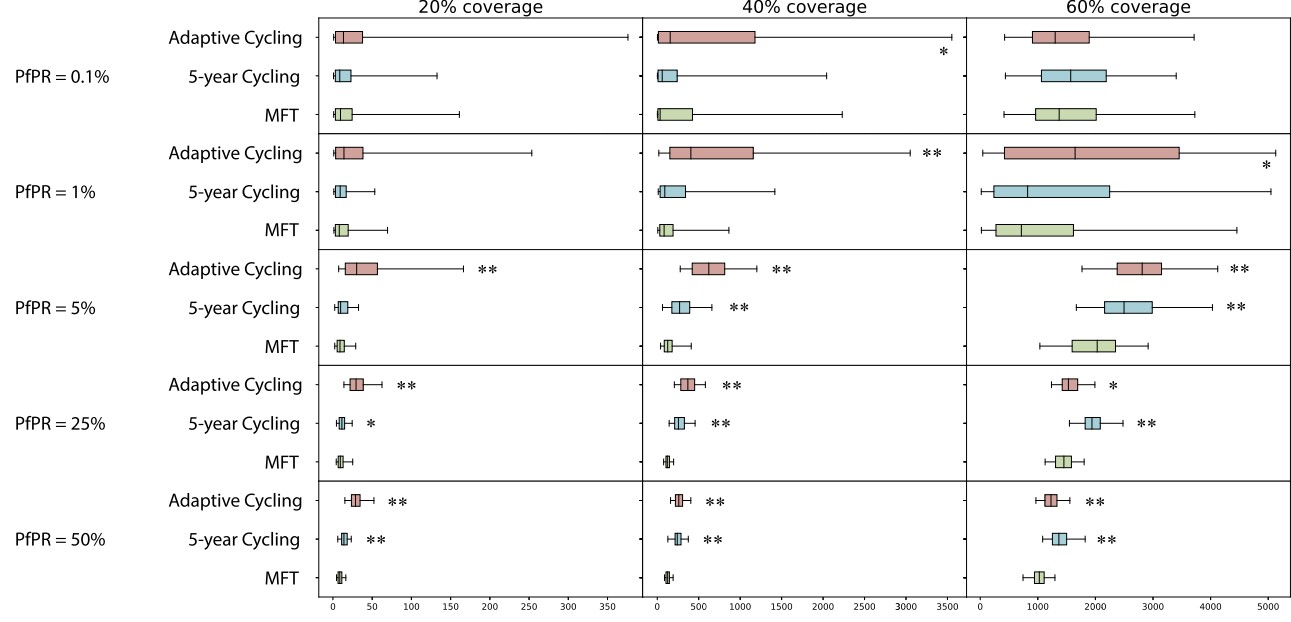

**Fig. 5 | Absolute risk of multi-drug resistance under an importation scenario over 20 years.** Each boxplot ($N = 100$ simulations) shows the sum of AUCs across all five maximally-resistant genotypes (Table 1). Boxplot whiskers are 1.5 times the IQR. AUC comparisons between MFT and 5-year cycling and AUC comparisons between MFT and adaptive cycling are assessed with a Mann-Whitney test, and $p$-value markers (testing whether MFT has lower AUC) are placed next to each boxplot with $p < 0.05$ (*) or $p < 10^{-4}$ (**). In these scenarios resistant genotypes are imported according to a Poisson process with a mean importation rate of one parasite per year (in an asymptomatic individual) with an equal 0.20 probability that the imported genotype is one of the five maximally-resistant parasites from Table 1.

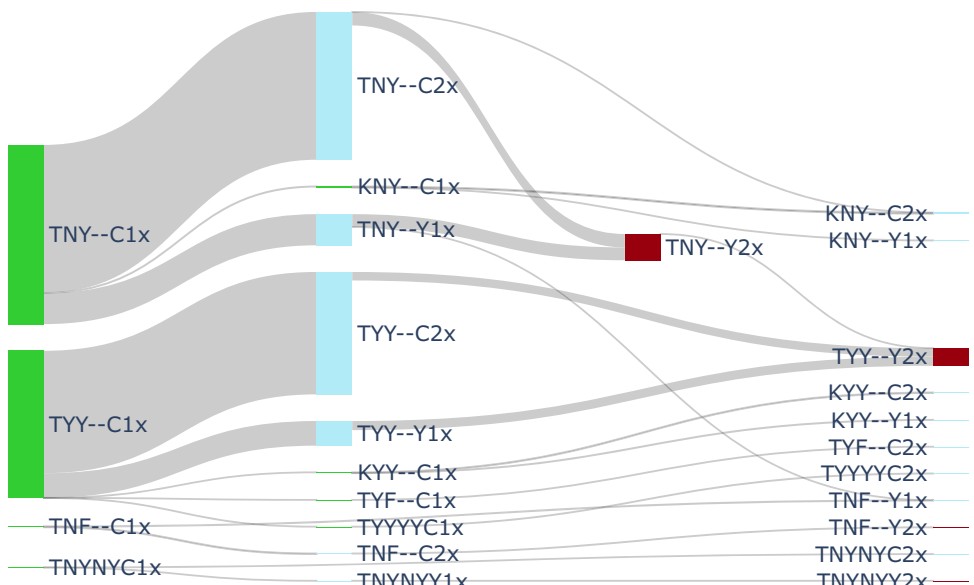

**Fig. 6 | Example of mutation flow during years 16 to 20 of a 5-year cycling strategy, where DHA-PPQ is used first, ASAQ second, and AL last.** PfPR$_{2-10}$ is 5% and treatment coverage is 40%. The diagram shows mutation flow during the second period of DHA-PPQ usage for a 'median' simulation. The median simulation was chosen by minimizing the absolute distance (among 100 runs) to the five median frequency lines shown in Fig. 2. Drug-sensitive genotypes are shown in green, single-resistant genotypes in blue, and the double-resistant to DHA-PPQ is shown in crimson ("2" connotes PPQ-resistance). Mutation occurs from left to right, and the width of the flow is proportional to the absolute number of mutations during the five-year period. A total of 372 mutations to the maximally-resistant triple-resistant (TYY--Y2) occur in years 16–20 of a 5-year cycling policy while the corresponding number of mutations for an MFT policy is 318. The total number of mutations to DHA-PPQ-AQ triple-resistant over 20 years is 581 for 5-year cycling (this figure) and 498 for MFT. Mutations shown are mutations that emerge and fix within host. Recombination occurs in the model but recombination events are not shown in the diagram.

The results cannot be extrapolated to extremely low-coverage settings (e.g. no drugs deployed or no access to treatment), high-coverage locations where nearly everyone is treated, or other scenarios that fall outside the bounds of what was studied.

Second, some parameter estimates come with substantial uncertainty, with the emergence rate of resistant genotypes—either de novo or through introduction via immigration—estimated from one of the most data-poor areas of drug-resistance epidemiology. Occasionally, the rate of appearance of certain mutants de novo is measured in clinical trial settings with limited numbers of patients[40], but these rates are not predictive of how quickly drug-resistance will emerge in populations of millions of individuals over years or decades. Likewise, the immigration rate of malaria genotypes from one country to another is not something that can be easily measured even with intensive genomic surveillance. For this reason, we calibrate our mutation rate to the approximate 5-10 year window in Cambodia (1980s to 1990s) that it would have taken artemisinin-resistant alleles to progress from initial mutation to 0.01 allele frequency[4,26]. A mutation rate ten times higher is implausible as it results in months-long fixation dynamics of drug resistance, something that has never been observed in the field. A mutation rate ten times lower results in (i) historically impossible emergence times on a multi-decade scale, and (ii) lack of emergence of MDR types in a 20-year time frame. We tested scenarios with 3-fold and 5-fold reduced mutation rates, but the main feature of these scenarios was that for most simulations multi-drug resistance did not emerge during the 20-year simulation window (Supplementary Figs. 35 and 36). Similarly, immigration rates would need to be custom-calculated for each epidemiological scenario based on proximity to a source of drug resistance. We present a 'rare migration' scenario in Fig. 5, but it is not guaranteed that MFT will remain the best policy option under continuous importation of drug resistance.

Third, one of the biggest challenges in drug-resistance modeling is estimating the partial extent of partial drug resistance. For $m$ possible therapies that could be deployed and $n$ loci that could improve a parasite's in vivo survival when exposed to the drug, we would need $m$ times $2^n$ estimates of partial efficacy of a particular therapy on a particular genotype. Clinical trials and therapeutic efficacy studies are not powered to estimate genotype-specific efficacies of antimalarial drugs[41], and regardless, not all genotypes are observed in nature and there are too few trials to enroll patients carrying all circulating genotypes. For this exercise, we have chosen 64 genotypes defined by six commonly sequenced loci and the three most widely used ACTs. These $3 \times 64 = 192$ genotype-treatment combinations represent the most data-rich region of drug-genotype space, and approximations of treatment efficacy can be made[24]. However, sensitivity analyses on this set of 192 treatment efficacies still show that we should expect substantial variation in emergence times (see Supplementary Section 4 here and Supplementary Section 5 in Watson et al.[26]). Until a better understanding of the genetic landscape of drug-resistant genotypes is achieved, there will continue to be substantial uncertainty as to which alleles and genotypes will emerge first.

When compared to cycling policies, simultaneous deployment of multiple first-line antimalarials creates drug environments with higher variability, delaying the emergence and slowing the evolution of drug-resistant genotypes and multi-drug resistant genotypes alike. This diversity principle will be critical for both the introduction of new antimalarials later in the decade and current response strategies[42] to the emergence of artemisinin resistance in Rwanda and Uganda. National Malaria Control Programs in many African countries will need to be prepared for the possibility of continuous monitoring and continuous adjustment of drug deployment strategies, as reversing artemisinin resistance will be a delicate task for as long as all approved therapies contain an artemisinin derivative. The reason that both MFT and adaptive MFT approaches should be seriously considered in the current context is that switching or rotation strategies are likely to be sub-optimal in slowing down the spread of artemisinin-resistant alleles. Given the pace of resistance spread seen so far, we will likely have only one opportunity to control this particular epidemic of drug resistance.

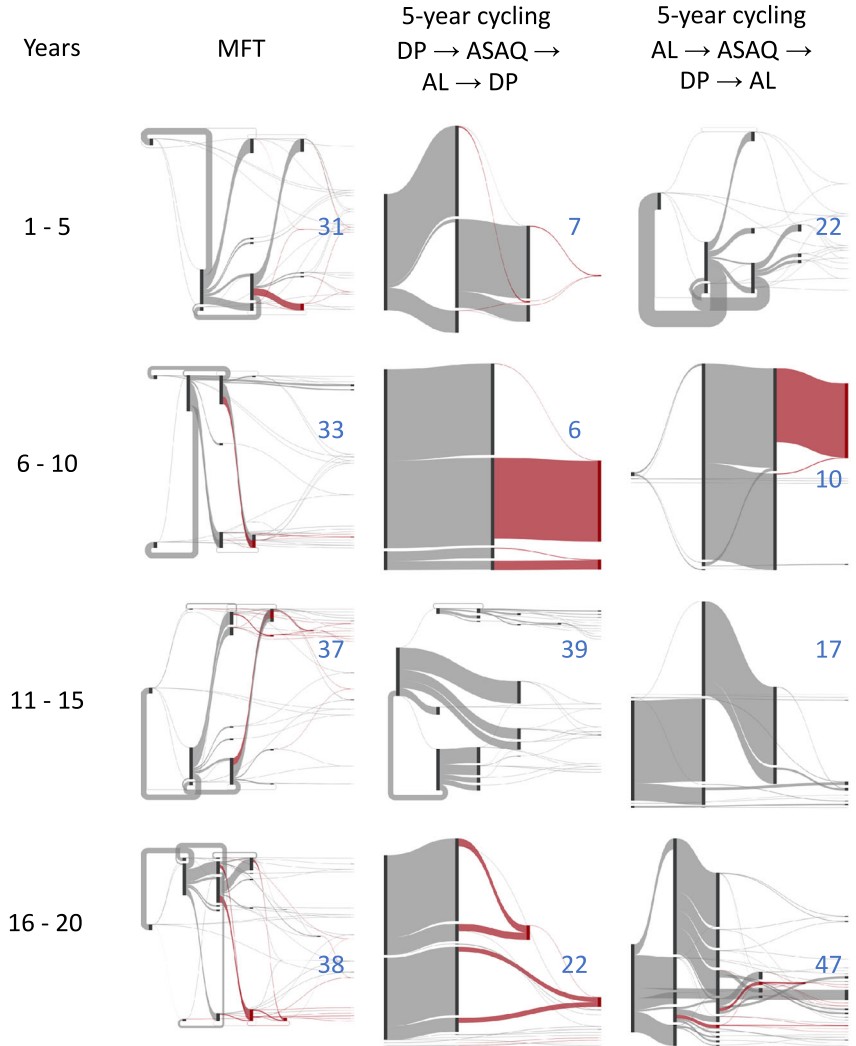

**Fig. 7 | Mutation flow diagrams for median MFT and 5-year cycling simulations, separated into four 5-year periods.** PfPR$_{2-10}$ is 5% and treatment coverage is 40%. Two different 5-year cycling strategies are explored (middle and right columns) with different ordering of ACT deployment. Mutation occurs left to right in the diagrams (but the *x*-axis here is not time) and the width of the flow is proportional to the absolute number of mutations occurring during the five-year period. The crimson-colored flows show evolution towards the maximally-resistant double-resistant genotypes (bottom three rows of Table 1). All other flows are shown in gray. The blue numbers in each panel show the number of 'destination genotypes' for the mutation and within-host-selection process; in other words these are the genotypes being selected for during each period. MFT shows more diversifying selection while 5-year cycling shows more unidirectional selection.

## Methods

We use a stochastic individual-based genotype-explicit pathogen transmission model whose core components were specifically developed for *Plasmodium falciparum* transmission and evolution[20,26,38]. All simulation details, data sources, and validations are described in previous publications[20,26,38] with the exact parameterization from Zupko et al.[38] used for the present analysis (but with one spatial location, no private-market drug use, and a population age-structure based on Tanzania). Briefly, the simulation has a daily time-step, at the beginning of which a Poisson number of individuals are chosen to be bitten by mosquitoes carrying particular genotypes of *P. falciparum*, with the genotype-specific force of infection (FOI) dependent on (1) the number of individuals carrying a particular genotype, (2) the individuals' biting attractiveness, (3) the parasite density within the individual, and (4) the probability that recombination generates a particular genotype based on the frequency and parasite density of currently circulating genotypes (using a traditional population-genetic recombination table). Individuals acquire and lose immunity at a previously calibrated rate (see Figures S3 to S6 in Nguyen et al.[20]), and symptoms presentation is dependent on each host's level of malaria immunity, which

is dependent on frequency and recency of past infections. Pharmacodynamics is modeled with a traditional Hill equation[43] and pharmacokinetics is modeled as 1-compartment clearance for all drugs.

The evolutionary and epidemiological aspects of the simulation are calibrated to (1) the relationship between prevalence in 2-10 year-olds (PfPR$_{2-10}$) and the entomological inoculation rate (EIR) for *P. falciparum*[38], (2) the relationship between age-specific incidence and EIR[20], (3) the relationship between EIR and multi-clonality of infections[20], (4) an approximate mutation rate based on the emergence pattern of artemisinin resistance[26], and (5) a genotype-environment interaction matrix based on clinical trial data[24]. The mutation rate is calibrated so that 40% treatment coverage of DHA-PPQ (and no private market drug use) in a 10% PfPR$_{all-ages}$ setting of 100,000 individuals produces an 0.01 allele frequency of the 580Y allele in exactly 7.0 years. This process is the most difficult to calibrate as the per-infection "mutation and within-host selection" probability is the most difficult to measure[26,44], but the current parametrization is within an order of magnitude of values that would replicate real-world timings of drug-resistance emergence in Cambodia. A burn-in phase of the simulation is started with one million individuals, and 10% are

randomly infected with one of two *P. falciparum* genotypes—either wild-type allele N86 at the *pfmdr1* locus or mutant allele 86Y, equally; remaining alleles are all wild-type except for the 76T locus at *pfcrt* (see below). The burn-in phase is run for ten years with 50% drug coverage of an 80% efficacious therapy, at which point the population reaches its equilibrium $PfPR_{2-10}$ level of 0.1%, 1%, 5%, 25%, or 50% (these were pre-calibrated). Once at equilibrium, artemisinin-combination therapies are introduced and the simulation is run for another twenty years. Note that when ACT coverage is 20% or 40%, prevalence may go up as fewer individuals would be receiving treatment than during burn-in.

## Locus structure

As in the most recent version of this model[24,26,38], we use a 6-locus structure focused on the key drug-resistance determining alleles for artemisinin (ART), lumefantrine (LUM), amodiaquine (AQ), and piperaquine (PPQ). Included loci are *pfcrt* K76T, *pfmdr1* N86Y, *pfmdr1* Y184F, *pfkelch13* C580Y, double-copy variant at *pfmdr1*, and a generic piperaquine-resistance locus determined primarily by a group of mutations in *pfcrt*[17,45–50]. The efficacy of different therapies on different genotypes was calibrated for a previous publication[24]. Briefly, for each of these 64 genotypes and for all four drugs modeled here, a specific EC50-value is assigned (calibrated) to each of the 256 drug-genotype combinations, allowing for 192 genotype-specific efficacies to be computed for AL, ASAQ, and DHA-PPQ. The EC50 values are translated into 28-day efficacies via a one-compartment pharmacokinetic model and daily parasite kill rate modeled via a standard Hill equation (see Nguyen et al.[20], supplement, section 9). The EC50 values are then calibrated so that the 28-day efficacies match cure rates from randomized controlled trials or therapeutic efficacy studies for the three different ACTs, with genotype information obtained from trial data, routine molecular surveillance, or historical context (depending on availability). For the current model version, we ignore the individual effects of the *pfcrt* mutations relevant to PPQ-resistance and assume that these emerge as a group or haplotype resulting in the lowest possible DHA-PPQ efficacies on the DHA-PPQ double-resistant genotype (around 41.5%). Finally, we use the term lumefantrine-resistant to describe alleles (K76, N86, 184F) that have been associated with reduced lumefantrine susceptibility[51–54] although high-grade lumefantrine resistance has not yet been observed in the field.

Using these loci we make specific definitions of double-resistant and triple-resistant genotypes, and these always have to be defined with respect to a particular therapy. In Table 1, we define the maximally-resistant double-resistants to AL, ASAQ, and DHA-PPQ, and we define the maximally-resistant triple-resistants to DHA-PPQ-LUM and DHA-PPQ-AQ, in the context of our 6-locus framework. Note that it is not possible to define a triple-resistant to AL-AQ because the pleiotropic effects at the *pfmdr1* and *pfcrt* genes prevent any of our modeled genotypes from being simultaneously resistant to amodiaquine and lumefantrine. The multi-genic resistant types in Table 1 may arise via recombination or successive mutations. The genotypes in Table 1 were chosen as they carry the most commonly observed resistance mutations to the most widely deployed ACTs.

For some model scenarios, genotypes with lower resistance levels are tracked. For example, the 580Y 184F genotype is a double-resistant to AL, but it does not have the full complement of lumefantrine resistance mutations (N86, 184F, K76). We call this genotype a double-resistant double-mutant, or a "2-2" genotype. In some cases (e.g. Fig. 1E, G for the AL resistant) the "2-2" genotypes are tracked if the maximally-resistant "2-4" genotypes do not emerge. In the color schemes in Figs. 1 to 3, the lighter red colors represent "2-2" genotypes and the darker red (crimson) colors represent "2-4" genotypes; in other words, darker red corresponds to stronger drug resistance.

## Treatment Strategies and Scenarios

At the population's natural endemic malaria equilibrium (modelled to mimic the chloroquine and SP era of treatment in Africa that lasted until 2005), artemisinin-combination therapies are introduced into use into one of three ways: (1) as the current status quo approach of recommending a single first-line ACT, which is rotated out when treatment failure with this ACT crosses the 10% failure threshold, with a one-year delay included in the rotation as changing first-line therapies cannot be implemented instantaneously by a large national health system; (2) as a five-year cycling approach where ACTs are rotated out and replaced every five years; and (3) as multiple first-line therapies (MFT) where all three therapies—DHA-PPQ, ASAQ, and AL—are deployed simultaneously in equal proportions. The order of the deployed therapies in cycling strategies is DHA-PPQ first, ASAQ second, and AL last, as this has the largest effect on prevalence reductions due to the longer half-life drugs being used first. A reverse-order cycling approach is analyzed in Fig. 7 to examine differences in the pattern of mutation and selection.

Fifteen epidemiological scenarios are evaluated. The simulation's transmission parameter, which controls the daily biting rate, was calibrated to achieve five different prevalence levels (0.1%, 1%, 5%, 25%, 50%) at three different drug coverage levels (20%, 40%, 60%). Drug coverage is the percentage of symptomatic falciparum cases that have access to treatment, seek treatment, and complete a full 3-day ACT course. In order to remove the effect of de novo mutation which has stronger effects at higher prevalence levels where there are higher absolute numbers of patients treated, the scenarios were re-run with high migration levels with a one-year waiting time (Poisson process) to the next immigration event of a maximally-resistant genotype, which is chosen at random with 1/5 probability from Table 1. Alternatively, this can be viewed as five independent Poisson processes for the five maximally-resistant genotypes, with a five-year waiting times for each genotype.

## Outcomes measures

In order to evaluate whether MFT will generate more or less multi-drug resistance than either cycling approach we introduce two key metrics for comparison. First, we track the time until a maximally-resistant genotype reaches 0.01 genotype frequency ($T_{.01}$). The 0.01 threshold is chosen as this frequency is high enough that the parasites will not be at risk of extinction due to the effects of random genetic drift, but low enough that the major evolutionary driving forces in reaching this threshold will be mutation, recombination, and within-host selection (see Supplementary Fig. 3 in Watson et al.[26]). A desirable outcome in drug-resistance management is that a chosen strategy is late at generating and establishing a lineage of multi-drug resistant parasites. Visually, we also show $T_{.10}$ as this is the point of establishment. The genotype frequency after 20 years ($x_{20}$) is also shown.

Second, we track the total number of "multi-drug resistant risk days" over a 20-year period. This is the area under the frequency curve (AUC) for each maximally-resistant genotype. This is the total risk generated by cumulative relative exposure to a particular maximally-resistant genotype, but it does not account for lower prevalence. In other words, this AUC measure characterizes the total evolutionary pressure for multi-drug resistance, but not the combined epidemiological-evolutionary pressure. This is a moot point in our analysis as prevalence values are nearly identical for MFT and cycling policies (Supplementary Fig. 37). A total AUC measure across all five maximally-resistant genotypes is also presented (Figs. 4 and 5), and note that the five individual-genotype AUCs are not simply added to obtain this total AUC measure. The double-resistant AUC quantities include the genotype frequencies of triple-resistant genotypes, and this double counting is removed when presenting the combined five-genotype AUC in Figs. 4 and 5.

Results are presented as medians, interquartile ranges (shaded regions), and 95% ranges (where specified) from 100 simulations. Differences between distributions are assessed with Mann-Whitney tests. Mutation flow diagrams were generation with Python library plotly v5.9.0.

## Reporting summary

Further information on research design is available in the Nature Portfolio Reporting Summary linked to this article.

## Data availability

All simulation outputs, i.e. the simulated data generated for this study, are available[55] at https://github.com/bonilab/malariaibm-generation-of-MDR-mutants.

## Code availability

All code is available[55] at https://github.com/bonilab/malariaibm-generation-of-MDR-mutants.

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

## Acknowledgements
EZL was funded by the University of Washington's Malaria Modeling Consortium grant from the Bill and Melinda Gates Foundation (OPP159934) and by Bill and Melinda Gates Foundation grant INV-005517. TDN, TN-AT, RJZ, MFB are funded by National Institutes of Health grant NIAID R01AI153355 and Bill and Melinda Gates Foundation grant INV-005517. Computations for this research were performed on the Pennsylvania State University's Institute for Computational and Data Sciences' Roar supercomputer.

## Author contributions
EZL, TDN, MFB designed the study. EZL, RJZ, TDN upgraded the simulation to explicitly and flexibly handle outputs on multi-drug resistant genotypes. EZL and TDN performed all simulations and EZL made all figures. EZL and TNAT analyzed evolutionary outcomes and verified that they were consistent with treatment outcomes in clinical trial data. MFB wrote the first draft of the paper. All authors contributed to editing the manuscript.

## Competing interests
The authors declare no competing interests.
