## [Peer Review File · Nature Communications]

Assessing emergence risk of double-resistant and triple-resistant genotypes of *Plasmodium falciparum*Editorial Note: Parts of this Peer Review File have been redacted as indicated to remove third-party material where no permission to publish could be obtained.

REVIEWER COMMENTS

Reviewer #1 (Remarks to the Author):

In this manuscript the authors present results from simulations run using an individual-based transmission model to evaluate the impact of deployment of different ACTs a) as co-equal multiple first-line therapies, b) rotating in five-year cycling increments, or c) rotating in response to detection of 10% treatment failure on selecting for double-resistant or triple-resistant malaria parasites under different parasite prevalences and drug coverage scenarios. As I am not a modeler my review doesn't critique the modeling, but focuses on the epidemiological and evolutionary aspects of the analysis.

I am very appreciative of the need to predict the impact of different treatment policies on evolving drug resistance and to identify best practices for maintaining the efficacy of available drugs, while awaiting new options to become available. In this context I found the authors' presentation quite compelling. However, as I'm sure is a common critique for all modeling studies, I do have some concerns about some of the assumptions being made, the limited range of the variable parameters being tested, and some of the terminology used to communicate the findings. Many of my comments are intended to facilitate communication of this data to a reader like myself: someone with knowledge about drug resistance and pharmacokinetics, but less understanding about the validation of underlying assumptions and subtleties of different models.

1. It would be helpful to better clarify how the treatment failure rates associated with the different genotypes in Table 1 were determined. It took me forever to find this information at the end of the discussion, and even then it just directed me to another reference. This is one of the places where it most clearly indicates what is meant by resistance.

2. I found the use of the term "drug resistant" bit misleading in some cases. This felt particularly problematic for "lumefantrine resistance". Stable resistance to lumefantrine has never been demonstrated, and most definitely is not conferred by the K76, N86, 184F. The need for a simplified term is like "resistance" is understandable, but using your term "maximally-resistant genotypes" seems preferable, or "reduced sensitivity" for cases where the genotypes are not clearly associated with true resistance phenotypes.

3. The role of 184F in decreased lumefantrine sensitivity is a bit unclear in the literature. It may only have a compensatory/fitness role and definitely does not seem to be undergoing the same population-level selection with AL use that is seen for *mdr1* 86. How would the outcomes of your model change if only 86 were considered?

4. Does the fact that not all mutational pathways are allowed for in the model for AL mean that by definition that lumefantrine cannot achieve a 0.01 maximal double-resistance frequency? If so, is this simulation informative?

5. The authors comment that maximally-resistant genotypes in low transmission scenarios do not reach frequencies above 0.01 because "at low prevalence, there are fewer infections, fewer parasites, and fewer opportunities for mutation." Yet, single drug resistance appears to have first emerged in low transmission areas of SEA for SP, chloroquine, piperazine and artemisinins and for double drug resistance to DP. This statement seems to contradict this observation. Can the authors clarify?

6. A prevalence of 25% seems low for many parts of Africa, including Uganda where artemisinin resistance appears to be emerging and policy changes may need to be reconsidered. Could the authors elaborate on the likelihood of the trends seen at the reported prevalences to continue to hold at higher prevalences, where recombination may be an even bigger threat?

7. For figure 6, when discussing the acquisition of additional mutations, do the authors mean de novo acquisition of each mutation? It's a complicated figure, but I didn't see an option for recombination. Is this because it was not considered as a possibility or because the model found that it was not a likely occurrence.

8. How would the emergence of novel mutations that strongly reduced the efficacy of AL impact the model? Lumefantrine has been a very good drug, and while it is possible it will be very difficult to select resistance, the fact that we have not seen resistance yet might also be attributed to the fact that the drug has not been seen as a monotherapy. With artemisinins starting to fail in Africa, it is possible lumefantrine will be under greater selective pressure, which might facilitate the emergence of novel mutations that confer higher levels of resistance. I recognize that this is a hypothetical, and the goal of this modeling exercise is to evaluate what could be done right now to prevent the evolution of additional resistance, but I think it is likely to be an important scenario. It

would also help me to understand how the magnitude of selection for a mutation under drug pressure impacts on the different scenarios.

Minor points:

1. Add a reference for the following phrase on page 4: "Evaluations were performed with a previously calibrated individual-based simulation of Plasmodium falciparum transmission and drug-resistance evolution."
2. Consider defining the significance of the 0.01 genotype frequency (genotype unlikely to be lost randomly) earlier in the manuscript, rather than leaving it until the results section.
3. Typo on pg 3: "(MFT) is the next best option as this forces the parasite population to encounter different a different ACTs every several weeks"
4. Typo on page 16: "This is a moot point in our analysis as prevalence values are nearly identical MFT and cycling policies"

Reviewer #2 (Remarks to the Author):

This manuscript uses simulations to test the effects of different drug usage strategies on the evolution of drug resistance in malaria. The authors find that using multiple drugs simultaneously (at the population level---each patient still only gets one drug) generally leads to slower resistance evolution than using a single drug at a time and cycling through them. This is an important, timely topic, and as the manuscript points out we need to use models to guide strategies right now and cannot just wait for the results of field trials. Again as the manuscript points out though, the big question with simulations is how well they will describe real-world outcomes. The manuscript addresses one side of this problem by considering the uncertainties in different model parameters. My main suggestion would also be to consider the flip side of the problem and conduct some form of sensitivity analysis to see which parameters create a lot of variability in the outcomes within their plausible ranges. A full simulation-based sensitivity analysis is a very big, unreasonable undertaking, but trying variations of a few key parameters (mutation and recombination perhaps?) for one key scenario would be very illuminating. The authors have already done something along these lines by testing the effects of having drug resistance be introduced by immigration rather than mutation.

Minor/typos/etc:

- p3: presumably "23 countries *with endemic malaria*" or something like that
 - p3: "different a different"
 - p5: "a genotype with two copies of the pfmdr1 gene is not allowed to mutate identically in both copies": confusing, better to write "a genotype with two copies of the pfmdr1 gene acquire two independent mutations to change both copies to 184F" or similar
 - Fig 3, 2nd row: y-axis shouldn't go negative
 - Fig 4: I found this hard to parse. It's kind of confusing that it's the boxes with *no* writing in them that we should pay the most attention to, if i understand correctly.
 - Fig 7: Another one that I found hard to understand. Why don't the lines look like they would flow into each other as you go from time period to time period? Time is the horizontal axis within each panel but the vertical axis between panels. I think the figure would be easier to follow if one of them were rotated so that time was consistently vertical or horizontal.
 - p11: "partial extent or partial drug resistance" is confusing
- Daniel Weissman

Dear Editors and Referees

Thank you for the feedback.

We apologize for the amount of time it has taken us to put this revision together. Simply put, the additional simulations at high prevalence (these are particularly computationally expensive) and the additional sensitivity analyses requested in a few places in the referee reports required us to spend most of the summer on preparing and running these scenarios. Most of the simulation work was done by early fall, and we spent the past two months on the assembly and write-up of the results.

Thank you for your patience. The referee comments were very constructive and helpful.

Best Wishes

Maciej Boni, on behalf of all authors

Reviewer #1 (Remarks to the Author):

In this manuscript the authors present results from simulations run using an individual-based transmission model to evaluate the impact of deployment of different ACTs a) as co-equal multiple first-line therapies, b) rotating in five-year cycling increments, or c) rotating in response to detection of 10% treatment failure on selecting for double-resistant or triple-resistant malaria parasites under different parasite prevalences and drug coverage scenarios. As I am not a modeler my review doesn't critique the modeling, but focuses on the epidemiological and evolutionary aspects of the analysis.

I am very appreciative of the need to predict the impact of different treatment policies on evolving drug resistance and to identify best practices for maintaining the efficacy of available drugs, while awaiting new options to become available. In this context I found the authors' presentation quite compelling.

However, as I'm sure is a common critique for all modeling studies, I do have some concerns about some of the assumptions being made, the limited range of the variable parameters being tested, and some of the terminology used to communicate the findings. Many of my comments are intended to facilitate communication of this data to a reader like myself: someone with knowledge about drug resistance and pharmacokinetics, but less understanding about the validation of underlying assumptions and subtleties of different models.

1. It would be helpful to better clarify how the treatment failure rates associated with the different genotypes in Table 1 were determined. It took me forever to find this information at the end of the discussion, and even then it just directed me to another reference. This is one of the places where it most clearly indicates what is meant by resistance.

This was done for a previous publication (PLoS Global Public Health 2023), and we summarize briefly now (section 4.1) what the calibration process looked like, and include the reference directly. The reference material here is quite long so we feel it's best to simply refer the reader directly; we are working on an updated version of these treatment failure rates and how they associate with different genotypes, but this will probably not be ready until 2025.

2. I found the use of the term “drug resistant” bit misleading in some cases. This felt particularly problematic for “lumefantrine resistance”. Stable resistance to lumefantrine has never been demonstrated, and most definitely is not conferred by the K76, N86, 184F. The need for a simplified term is like “resistance” is understandable, but using your term “maximally-resistant genotypes” seems preferable, or “reduced sensitivity” for cases where the genotypes are not clearly associated with true resistance phenotypes.

Yes, the referee is right that the term “maximally-resistant genotypes” is misleading when viewed in isolation. We have modified our explanation the first time this term appears to make clear that this is in no way a statement on current or future lumefantrine resistance (see new results section, para 2).

The discussion on whether the observed reduced sensitivity to lumefantrine should be termed “reduced sensitivity” or “resistance” or “partial resistance” is one we are very familiar with. We have added a comment in the methods (para 3) to this effect.

In our opinion, variations in lumefantrine susceptibility can be viewed as partial lumefantrine resistance. People do have opinions on what the correct terminology is, but this is a matter of semantics as what really matters is the 28-day efficacy of artemether-lumefantrine (AL) because this is how we judge whether we have noticeable resistance that leads to treatment failures.

There are four analyses (that we know of) with meaningful data on how LUM efficacy or AL efficacy varies by genotype, and these studies tell us that there is some genetic basis for reduced lumefantrine sensitivity (or partial resistance). They are

1. Baraka et al 2015. “In Vivo Selection of Plasmodium Falciparum Pfcrf and Pfmdr1 Variants by Artemether-Lumefantrine and Dihydroartemisinin-Piperaquine in Burkina Faso.” *Antimicrobial Agents and Chemotherapy* 59 (1): 734–37. <https://doi.org/10.1128/AAC.03647-14>.
2. Bassat et al. 2009. “Dihydroartemisinin-Piperaquine and Artemether-Lumefantrine for Treating Uncomplicated Malaria in African Children: A Randomised, Non-Inferiority Trial.” *PLoS One* 4 (11): e7871. <https://doi.org/10.1371/journal.pone.0007871>.
3. Kiaco et al. 2015. “Evaluation of Artemether-Lumefantrine Efficacy in the Treatment of Uncomplicated Malaria and Its Association with Pfmdr1, Pfatpase6 and K13-Propeller Polymorphisms in Luanda, Angola.” *Malaria Journal* 14 (1). <https://doi.org/10.1186/s12936-015-1018-3>.
4. Plucinski et al. 2015. “Efficacy of Artemether-Lumefantrine and Dihydroartemisinin-Piperaquine for Treatment of Uncomplicated Malaria in Children in Zaire and Uíge Provinces, Angola.” *Antimicrobial Agents and Chemotherapy* 59 (1): 437–43. <https://doi.org/10.1128/AAC.04181-1>

According to the efficacy data in these studies (which can be broken up by genotype), the N86Y and Y184F loci in pfmdr1, and the K76T locus in pfcrf are weakly predictive of LUM and AQ resistance. What this means is that efficacies can vary from 95% to 90%, and sometimes down to 85% or 80% when all alleles are of the ‘partially resistant to lumefantrine’ type (K76 N86 184F). The effect sizes are small and the data do vary among studies. But the evidence does point to a genetic basis for small increases in treatment failure probability resulting from lower falciparum sensitivity to lumefantrine.

In late October 2023, at a WHO Malaria Policy meeting, these AL efficacy data were presented for Uganda (meaning that they are now in the public domain):

[REDACTED]

AL efficacy has dropped to 82% in two sites where ASAQ efficacy and Pyr-AS efficacy are still above 90%. Hopefully, over the next 12 months, this will make the malaria clinical and policy communities more comfortable with using the term “lumefantrine resistance”.

3. The role of 184F in decreased lumefantrine sensitivity is a bit unclear in the literature. It may only have a compensatory/fitness role and definitely does not seem to be undergoing the same population-level selection with AL use that is seen for *mdr1* 86. How would the outcomes of your model change if only 86 were considered?

Yes, the referee is correct here. This is important. In addition, copy number of *pfmdr1* also appears to have a weak effect or no effect (with more recent data available) on lumefantrine sensitivity.

We added a new analysis, where we generated 3 new drug-by-genotype efficacy tables. These tables were

1. Original table, but with effect of Y184F on lumefantrine zeroed out (i.e. we treat this as a neutral allele with respect to lumefantrine sensitivity/resistance)
2. Original table, but with effect of copy number variation (CNV) of *pfmdr1* zeroed out
3. Original table, but with both effects of Y184F and CNV of *pfmdr1* zeroed out for lumefantrine

The overall AUC values (what we use to measure MDR risk) and the general relationships among the strategies and their associated MDR risk did not change. Detailed in section 4 of the supplement.

4. Does the fact that not all mutational pathways are allowed for in the model for AL mean that by definition that lumefantrine cannot achieve a 0.01 maximal double-resistance frequency? If so, is this simulation informative?

This was poorly phrased by us and has been edited. All genotypes can indeed appear in the model, through a simple process of successive mutations (as everything in evolutionary biology and life as we know it). We have added a bit more detail here about why lumefantrine resistance evolution is slow in our models (and in real life).

See paragraph 3 of the results.

Examples of mutational pathways that are not allowed are

KNYNY1 cannot mutate to KNFNFY1

KYFYFY1 cannot mutate to KNFNFY1

As these would both require double-mutations to occur. A wild-type parasite can mutate to the two genotypes above, but only through pathways that proceed by single mutational steps.

5. The authors comment that maximally-resistant genotypes in low transmission scenarios do not reach frequencies above 0.01 because “at low prevalence, there are fewer infections, fewer parasites, and fewer opportunities for mutation.” Yet, single drug resistance appears to have first emerged in low transmission areas of SEA for SP, chloroquine, piperaquine and artemisinins and for double drug resistance to DP. This statement seems to contradict this observation. Can the authors clarify?

Yes, good question. We are worried about opening up a can of worms with this topic, so we just briefly commented on it in the main text (para 6 of the results) as it is a bit out of scope for this paper. A fuller explanation is below.

A lot has been published on this topic (we have included 4 key citations in para 6 that provide good summaries).

First, people tend to ignore ‘drug coverage’ or ‘drug access’ as a variable when analyzing why low-transmission regions tend to generate drug-resistance earlier or more often than high-transmission regions. Low transmission regions tend to be low transmission because drug access is high, and high levels of drug access (specifically, a high percentage of symptomatic malaria cases treated with a complete course of a high-efficacy treatment) also put a lot of selection pressure on the parasites to evolve resistance. A high-transmission region with high drug access (e.g. Uganda) should also be able to produce drug-resistant parasites.

Second, the two major reasons cited for low-transmission areas generating more drug resistance are (1) higher probability of symptoms progression, due to lower immunity, and thus higher probability of treatment, and (2) absence of within-host competition that favors the wild types. These are good arguments. However, these two forces are still likely weaker than the drug access/drug coverage variable in the paragraph above. See new supplementary figures 5 and 6 in the revised submission, where this comparison can be made.

Third, it is tautologically true that high-transmission regions produce more mutations, even if these mutations are lost due to absence of positive selection forces (little drug use) or presence of negative selection forces (within-host competition). The simple reason that there are few MDR mutants in figure 3 is that these simulations represent a population of one million individuals with 0.1% prevalence, meaning a total of 1000 infected individuals at any one time. These 1000 infections produce novel mutations at a slower rate than 500,000 infected individuals in a 50% PfPR region.

6. A prevalence of 25% seems low for many parts of Africa, including Uganda where artemisinin resistance appears to be emerging and policy changes may need to be reconsidered. Could the authors elaborate on the likelihood of the trends seen at the reported prevalences to continue to hold at higher prevalences, where recombination may be an even bigger threat?

We added a 50% prevalence level to all analyses. Figures 4 and 5 show the summaries in the bottom row. There is no major qualitative change in MDR risk, even though recombination is occurring (in the model, under the hood) more frequently in these 50% PfPR scenarios.

We have done some specific analyses on how recombination affects resistance emergence, but these are incomplete (at the moment) and they are being done with a newer version of the simulation where recombinant processes can be tracked more explicitly. The results are not all in yet, and we are not sure yet how strong of an effect recombination plays in these high-transmission scenarios.

(high-prevalence runs are slow in our simulation, as nearly each person and each clonal population needs to be updated in the simulation's routine update algorithms ... the calibration and scenario evaluation for this 50% PfPR region was done from May to July, and remaining analyses were done after this calibration was complete ... this is one of the reasons that the revision was submitted so late).

7. For figure 6, when discussing the acquisition of additional mutations, do the authors mean de novo acquisition of each mutation? It's a complicated figure, but I didn't see an option for recombination. Is this because it was not considered as a possibility or because the model found that it was not a likely occurrence.

Only mutations are counted in this figure, and specifically, mutations that appear within-host and then fix within-host. Recombination does occur in the model, but we cannot track and count individual recombination events in our model at the moment (see above); a newer version of our model will be able to do this.

The reason we cannot track individual recombination events is that this part of our simulation is set up like a traditional set of population-genetic equations with a recombination mating table. Population-level effects of recombination are included, but individual occurrences cannot be counted.

Figure 6 caption has been edited to clarify this point.

8. How would the emergence of novel mutations that strongly reduced the efficacy of AL impact the model? Lumefantrine has been a very good drug, and while it is possible it will be very difficult to select resistance, the fact that we have not seen resistance yet might also be attributed to the fact that the drug has not been seen as a monotherapy. With artemisinins starting to fail in Africa, it is possible lumefantrine will be under greater selective pressure, which might facilitate the emergence of novel mutations that confer higher levels of resistance. I recognize that this is a hypothetical, and the goal of this modeling exercise is to evaluate what could be done right now to prevent the evolution of additional resistance, but I think it is likely to be an important scenario. It would also help me to understand how the magnitude of selection for a mutation under drug pressure impacts on the different scenarios.

Very important question. We have been asked this in the reviews of previous manuscripts and finally decided to do an analysis of this hypothetical situation. This, also, was about a month's worth of parameterization and computational time.

Section 3 of the supplement shows this analysis. We evaluated novel lumefantrine resistance at modest, intermediate, and strong effects for this new hypothetical locus. As expected, with the introduction of this locus, MDR risk levels went up (Supp Figures 1 and 2). The relationship between MFT and Cycling did not change.

Minor points:

1. Add a reference for the following phrase on page 4: "Evaluations were performed with a previously calibrated individual-based simulation of Plasmodium falciparum transmission and drug-resistance evolution."

Done.

2. Consider defining the significance of the 0.01 genotype frequency (genotype unlikely to be lost randomly) earlier in the manuscript, rather than leaving it until the results section.

Good idea. This is now in para 2 of the results.

3. Typo on pg 3: "(MFT) is the next best option as this forces the parasite population to encounter different a different ACTs every several weeks"

Fixed, thank you.

4. Typo on page 16: "This is a moot point in our analysis as prevalence values are nearly identical MFT and cycling policies"

Fixed, thank you.

Reviewer #2 (Remarks to the Author):

This manuscript uses simulations to test the effects of different drug usage strategies on the evolution of drug resistance in malaria. The authors find that using multiple drugs simultaneously (at the population level---each patient still only gets one drug) generally leads to slower resistance evolution than using a single drug at a time and cycling through them. This is an important, timely topic, and as the manuscript points out we need to use models to guide strategies right now and cannot just wait for the results of field trials. Again as the manuscript points out though, the big question with simulations is how well they will describe real-world outcomes. The manuscript addresses one side of this problem by considering the uncertainties in different model parameters. My main suggestion would also be to consider the flip side of the problem and conduct some form of sensitivity analysis to see which parameters create a lot of variability in the outcomes within their plausible ranges. A full simulation-based sensitivity analysis is a very big, unreasonable undertaking, but trying variations of a few key parameters (mutation and recombination perhaps?) for one key scenario would be very illuminating. The authors have already done something along these lines by testing the effects of having drug resistance be introduced by immigration rather than mutation.

Very important comment, thank you.

For the revision, we have done a sensitivity analysis that is now included in section 5 of the supplement. Some of the associations were expected: higher treatment coverage and higher mutation rate lead to greater risk of multi-drug resistance. Longer cycling periods really do lead to higher MDR risk, indicating that it is likely successive mutations and not recombination that is generating MDR in these model runs (more work needs to be done here).

As a side benefit of this exercise, we can now see that drug coverage has a much stronger effect on MDR emergence than transmission level.

Unfortunately, we cannot add recombination into the sensitivity analysis. Our next model version (out in 2024) will have an interrupted feeding rate for mosquitoes and will allow us to vary the rate of recombination.

Minor/typos/etc:

- p3: presumably "23 countries *with endemic malaria*" or something like that

We changed this to "previously endemic". Three years of zero indigenous malaria cases is the WHO definition of national malaria elimination.

- p3: "different a different"

Fixed thank you.

- p5: "a genotype with two copies of the pfmdr1 gene is not allowed to mutate identically in both copies": confusing, better to write "a genotype with two copies of the pfmdr1 gene acquire two independent mutations to change both copies to 184F" or similar

Yes, this was confusing. Referee 1 commented on this as well. This section has been edited.

- Fig 3, 2nd row: y-axis shouldn't go negative

We were trying here to show in some clear way that these median genotype frequencies are zero throughout the entire 20-year simulation. We have kept this as is for now, as the other variations of this graph were less clear. If the referees and editors insist on changing this axis to a certain set of specific values, we can do that.

- Fig 4: I found this hard to parse. It's kind of confusing that it's the boxes with *no* writing in them that we should pay the most attention to, if I understand correctly.

Good suggestion. We have changed Figures 4 and 5 so that stars indicate statistical significance.

- Fig 7: Another one that I found hard to understand. Why don't the lines look like they would flow into each other as you go from time period to time period? Time is the horizontal axis within each panel but the vertical axis between panels. I think the figure would be easier to follow if one of them were rotated so that time was consistently vertical or horizontal.

Time is not the horizontal axis on each panel. We have clarified this now. This is just a network diagram

- p11: "partial extent or partial drug resistance" is confusing

This was a typo. It is the partial extent of partial drug resistance. Fixed now.

- Daniel Weissman

REVIEWERS' COMMENTS

Reviewer #1 (Remarks to the Author):

I thank the authors for all the work they put into doing the additional simulations required to address my questions and for the addition of the clarifying statements. I found them very helpful and feel that all of my concerns have been addressed.

Reviewer #2 (Remarks to the Author):

The authors have thoroughly addressed my comments. I do still feel that Fig 7 is hard to read, but I understand that the authors really just want the reader to see the absence of big "block flows" in the left column.